# Gut Microbiome and Metabolites in Patients with NAFLD and after Bariatric Surgery: A Comprehensive Review

**DOI:** 10.3390/metabo11060353

**Published:** 2021-05-31

**Authors:** Jacqueline Hoozemans, Maurits de Brauw, Max Nieuwdorp, Victor Gerdes

**Affiliations:** 1Department of Internal and Vascular Medicine, Amsterdam University Medical Centers, AMC, 1105 AZ Amsterdam, The Netherlands; m.nieuwdorp@amsterdamumc.nl (M.N.); v.e.gerdes@amsterdamumc.nl (V.G.); 2Department of Bariatric and General Surgery, Spaarne Hospital, 2134 TM Hoofddorp, The Netherlands; mdebrauw@spaarnegasthuis.nl; 3Department of Internal Medicine, Spaarne Hospital, 2134 TM Hoofddorp, The Netherlands

**Keywords:** NAFLD, bariatric surgery, gut microbiome, metabolites

## Abstract

The prevalence of non-alcoholic fatty liver disease (NAFLD) is increasing, as are other manifestations of metabolic syndrome such as obesity and type 2 diabetes. NAFLD is currently the number one cause of chronic liver disease worldwide. The pathophysiology of NAFLD and disease progression is poorly understood. A potential contributing role for gut microbiome and metabolites in NAFLD is proposed. Currently, bariatric surgery is an effective therapy to prevent the progression of NAFLD and other manifestations of metabolic syndrome such as obesity and type 2 diabetes. This review provides an overview of gut microbiome composition and related metabolites in individuals with NAFLD and after bariatric surgery. Causality remains to be proven. Furthermore, the clinical effects of bariatric surgery on NAFLD are illustrated. Whether the gut microbiome and metabolites contribute to the metabolic improvement and improvement of NAFLD seen after bariatric surgery has not yet been proven. Future microbiome and metabolome research is necessary for elucidating the pathophysiology and underlying metabolic pathways and phenotypes and providing better methods for diagnostics, prognostics and surveillance to optimize clinical care.

## 1. Introduction

Non-alcoholic fatty liver disease (NAFLD) is considered the hepatic manifestation of metabolic syndrome, next to obesity, insulin resistance and hyperlipidemia. The prevalence of obesity is increasing worldwide, as is the prevalence of NAFLD [1]. The term NAFLD comprises a spectrum of liver diseases, ranging from hepatic steatosis to non-alcoholic steatohepatitis (NASH) with or without fibrosis [2]. Progression of NASH can ultimately lead to liver cirrhosis or hepatocellular carcinoma. Moreover, NAFLD is considered to become the number one cause of chronic liver disease worldwide, with end-stage fatty liver disease as the second indication for liver transplantation [3,4,5].

Consequently, health care costs of liver disease are increasing, including costs for regular follow-up necessary to detect disease progression. The early stages of fatty liver disease can be reversible with a healthy lifestyle and weight loss. Currently, bariatric surgery is a long-term effective therapy to prevent the progression of NAFLD and other manifestations of metabolic syndrome, such as obesity and type 2 diabetes [4,6].

However, the exact pathophysiology of fatty liver disease progression remains to be clarified. The gut microbiome has been proposed as one potential contributing factor in the pathogenesis of NAFLD. The gut microbiome consists of microorganisms present in the gastrointestinal tract. The bacteria produce metabolites which are transported to the liver, where they can act as signaling molecules and have systemic effects. In addition, several studies have shown an influence of the gut microbiome on metabolism and suggest a causal role in the pathogenesis of obesity, diabetes and atherosclerotic vascular disease, which are all associated with NAFLD [7,8,9]. However, causal relations were mostly demonstrated in animal models. Translating and reproducing insights to and establishing causality in humans remains challenging.

In this review, we summarize the current evidence in humans on the changes in the gut microbiome and microbial metabolites after bariatric surgery, with a focus on the impact of gut microbiome and metabolites on NAFLD (Figure 1).

## 2. Gut Microbiome and Metabolites

The microbiota consists of all host microorganisms, including bacteria, protozoa, viruses and bacteriophages. The genetic material of all microorganisms is called the microbiome. The microbiome can be differentiated by location, such as the gut microbiome, oral microbiome, dermal microbiome and vaginal microbiome [10].

The gut microbiome is shaped by environmental, dietary and host factors, such as gastro-intestinal anatomy and pH [11]. Mode of delivery influences the infant gut microbiome at birth [12]. Most gut microbiota are not (directly) pathogenic, and do not cause local or systemic infection [13].

The total bacteria count in the human body is approximately 4 × 10 ^12^, equal to the number of human cells [10]. The most abundant in the gut microbiome of average lean adults are the phyla *Bacteroidetes* and *Firmicutes*, followed by *Proteobacteria*, *Fusobacteria* and *Actinobacteria* [14]. The composition of the microbiome can be affected by diet, antibiotics, and other medication such as metformin and proton pump inhibitors [15].

One of the functions of the gut microbiome is to contribute to host metabolism and homeostasis. Therefore, a distinct balance between bacterial species should be maintained. Disturbance of this balance is recognized as dysbiosis and has been associated with gastrointestinal complaints and metabolic alterations. For example, in a comparison of compositions of the gut microbiome between those with normal glucose, impaired glucose, and a diabetic glucose control, differences were observed [16,17]. Methods to determine microbiome composition include 16S ribosomal RNA (16S RNA) sequencing of ubiquitous genes and whole genome shotgun sequencing of the entire gene component. Additional functional analysis of the microbiome can provide insight into the metabolic microbial capacity.

The gut microbiota are metabolically active. Ingested food is metabolized by specific microorganisms and gut microbiota are also able to excrete metabolites. These gut microbiome-derived metabolites are absorbed by enterocytes and enter the circulation where they are able to exert systemic effects (Figure 1). Some of these metabolites are thought to be the key components in influencing host metabolism [18]. Dysbiosis, the disbalance of the gut microbiome, alters (diet-derived) metabolite production and thus crosstalk between the host and gut microbiome. Research on gut microbiome-derived metabolites helps to distinguish which ones are key players in regulating metabolism at cell level [18,19].

A major class of intestinal bacteria produced metabolites are short-chain fatty acids (SCFAs). They are produced during the fermentative activity of microbiota when digesting dietary fiber, and SCFA production is suggested to modulate intestinal pH and intestinal barrier integrity [20,21,22]. The main SCFAs are propionate, butyrate and acetate, which also function as signaling molecules in immune response regulation, intestinal homeostasis and energy metabolism [23]. Other classes of metabolites include amino acids such as glycine and its precursor serine, aromatase amino acids (AAAs) including phenylalanine and tyrosine, and branched chain amino acids (BCAAs) such as leucine or valine. BCAAs promote glucose uptake and skeletal protein synthesis [24,25]. However, increased serum levels of BCAAs are seen in individuals with insulin resistance, and two bacterial species were identified as the main species driving the association between the biosynthesis of BCAAs and insulin resistance [26]. The relationship between the gut microbiome and glucose metabolism has been reviewed elsewhere [27,28].

Furthermore, the gut microbiome metabolizes dietary choline to trimethylamine (TMA), which is metabolized by the liver to trimethylamine N oxide (TMAO) [29]. Elevated TMAO has been associated with atherosclerotic vascular disease [30].

Gut microbiota conjugate primary bile acids, produced by hepatocytes, into secondary bile acids [31,32]. Bile salt hydrolase (BSH) is one of the microbial enzymes involved in driving the deconjugation of bile acids, with metabolites such as glycine and taurine as end products [33]. Besides direct involvement in fat digestion, secondary bile acids also function as signaling molecules in lipid, glucose and energy metabolism [34,35]. Most bile acids are actively reabsorbed from the intestinal lumen to the liver. This gut-to-liver axis plays a role in the enterohepatic circulation of bile acids, influencing host metabolism. The Farnesoid X receptor (FXR) is involved in bile acid synthesis control and enterohepatic circulation [36,37]. Bile acid-induced TGR5 activation decreases inflammation, and alterations of bile acid metabolism are associated with gut microbiota dysbiosis and obesity [33,38,39]. Furthermore, imbalance in bile acids is associated with gut barrier dysfunction [40].

Lipopolysaccharides (LPS) are endotoxins, components of the Gram-negative bacterial cell membrane, and play a role in immune response [41]. They can also be transported beyond the liver (that filters most of these) into the systemic circulation, where they dysregulate the inflammatory tone and contribute to metabolic disease [42].

The development of advanced sequencing techniques and the use of machine learning has increased the possibilities to gain insight in the human microbiome and derived metabolites and differences between specific populations, forming the foundation of a whole new scale of possible therapies [43,44,45]. The influence of the gut microbiome in diseases such as diabetes mellitus or inflammatory bowel disease has been intensely investigated [46,47]. Intervention studies with prebiotics, probiotics, antibiotics and fecal microbiota transplantations (FMT) focus on elucidating a potential causal relationship between gut microbiome and development as therapy for various human diseases as well.

## 3. Gut Microbiome and Metabolites in NAFLD

### 3.1. Fatty Liver Disease

As briefly mentioned before, NAFLD includes various forms of fatty liver disease and can progress to severe and irreversible liver disease (Figure 1). Fatty liver disease encompasses hepatic steatosis and non-alcoholic steatohepatitis (NASH). NAFLD is defined as lipid droplet accumulation in more than 5% of hepatocytes, and is sometimes referred to “simple” steatosis [2]. NASH, characterized by inflammation, includes steatohepatitis with and without fibrosis and requires a liver biopsy for diagnosis. NASH is histologically characterized by inflammation and hepatocyte ballooning in combination with steatosis. Histological evaluation of liver biopsies is performed via scoring systems to classify liver disease, such as the NAFLD Activity Score (NAS) and the Steatosis Activity and Fibrosis Score (SAF). Another frequently used score is the NASH Clinical Research Network (NASH CRN) scoring system: F0 = no fibrosis; F1 = perisinusoidal or portal/periportal fibrosis; F2 = perisinusoidal and portal/periportal fibrosis; F3 = bridging fibrosis; and F4 = cirrhosis. European guidelines recommend the SAF score because of its higher accuracy in distinguishing the intermediate category from mild and severe fatty liver disease [48].

It is important to realize that although NAFLD is seen as the hepatic manifestation of metabolic syndrome, not all patients with NAFLD are obese, and several patients with obesity do not have NAFLD either [49].

### 3.2. Gut Microbiome in Non-Alcoholic Fatty Liver Diesease

Differences in composition of the gut microbiome of patients with fatty liver disease have been observed (Table 1) [50,51]. Compared to individuals without NAFLD, fatty liver disease is associated with an increased abundance of Gram-negative microbiota: the abundance of the phylum *Proteobacteria* is increased [52,53]. The genera *Escherichia*, *Enterobacteriaceae, Dorea* and *Peptinophilus* are enriched, and abundances of *Anaerosporobacter*, *Coprococcus, Eubacterium,* and *Faecalibacterium* and *Prevotella*, *Rikenellaceae* and *Ruminococcaceae* are decreased compared to healthy controls [52,54,55,56]. Functional analyses of these microbiome differences reported increased microbial capacity for the metabolism of BCAAs and AAAs [52,54,55,56]. Furthermore, increased intestinal permeability, which is associated with dysbiosis, is seen in NAFLD [57]. Increased intestinal permeability and inflammation was observed in NAFLD patients in combination with dysbiosis of the gut microbiota: higher abundance of *Escherichia* was observed in fatty liver disease [58]. Development of fatty liver disease to NASH is associated with increased innate immune activation and inflammation [59,60]. The specific inflammatory factors associated with NAFLD are reviewed elsewhere [61].

Furthermore, differences between the gut microbiome of the different manifestations of fatty liver disease are also reported. One study comparing the gut microbiome found higher abundance of the phylum *Firmicutes* in patients with mild NAFLD compared to advanced fibrosis. The species *Eubacterium rectale* and *Bacteroides vulgatus* were the most abundant species in the mild NAFLD group. The advanced fibrosis group had higher abundance of the phylum *Proteobacteria* and, the most abundant species were *Bacteroides vulgatus* and *Escherichia coli*. *Ruminococcus obeum* CAG: 39, *R. obeum,* and *E. rectale* were significantly lower in advanced fibrosis than in mild/moderate NAFLD. A study including 203 individuals compared patients with biopsy-proven NAFLD–cirrhosis to individuals with fatty liver without advanced fibrosis, and to controls without fatty liver disease, as determined via imaging. An enriched abundance of *Streptococcus* was observed in both NAFLD groups. *Megasphaera* was only enriched in NAFLD–cirrhosis, and the highest abundance in this group was family *Enterobacteriaceae* and genera *Streptococci* and *Gallibacterium*. Patients with fatty liver disease without advanced fibrosis and controls had similar abundances of *Bacillus* and *Lactococcus*, which was enriched compared to NAFLD-cirrhosis. Controls had a higher abundance of *Faecalibacterium prausnitzii* species compared to both NAFLD groups [62]. A recent prospective study, where fatty liver status was assessed via transient elastography, found decreased relative abundance of *Clostridium* (sensu stricto) in patients with liver fibrosis, and an enrichment of *Enterobacteriaceae*, *Escherichia* and *Shigella* compared to individuals with severe steatosis without fibrosis [63].

A study comparing liver biopsies of 57 patients with NAFLD reported that increased abundance of the family *Bacteroidaceae* and decreased abundance of the families *Prevotellacea* and *Erysipelotrichaceae* is associated with increased disease severity. Compared to patients without NASH, NASH was associated with decreased *Prevotella* and increased *Bacteroides* abundance. Fatty liver disease with significant fibrosis (F3/F4) is associated with the increased abundance of *Ruminococcus* and *Bacteroides*, and increased abundance of *Prevotella* compared to mild fibrosis [64]. In children, high abundance of *Prevotella copri* is associated with more severe fibrosis [65]. Another study found that compared to controls, the genus *Collinsella* was most strongly associated with patients with NASH, and *Ruminococcoaceae*, a SCFA-producing genus, was decreased [66].

Certain strains, such as the Proteobacteria species *Klebsiella pneumonia*, are associated with endogenous alcohol production and are found to be increased in fecal microbiome of individuals with NAFLD and NASH, and abundance decreased with clinical improvement of the fatty liver disease [9]. In children, NASH was associated with an increased abundance of Proteobacteria and increased peripheral blood ethanol levels. Between obese children and children with NASH, the microbiome abundance was only statistically different for the abundance of Proteobacteria, Enterobacteriaceae and *Escherichia* [52].

As already mentioned, NAFLD can also develop in non-obese patients. Additionally, it appears that microbiome and metabolome signatures of non-obese patients with NAFLD differ from those of obese patients [67]; however, this requires further study.

### 3.3. Metabolites in Fatty Liver Disease

The liver plays a central role in metabolism and homeostasis, and the majority of blood to it is supplied via the portal vein. The portal vein drains a large part of the gastrointestinal tract, and thus the liver is the first organ to process absorbed components, such as gut microbiome-derived metabolites.

A cohort study involving obese, non-diabetic women, associated hepatic steatosis with microbiome-derived metabolite phenylacetic acid [29]. Furthermore, low microbiome gene richness was correlated with plasma BCAAs leucine, valine and isoleucine and hepatic steatosis, and the microbial capacity for metabolism of BCAA and AAAs such as phenylalanine, tyrosine and tryptophan was increased in patients with steatosis [29]. Plasma choline and phosphocholine were not found to be negatively correlated with steatosis, but urinary choline excretion was increased [29]. However, TMA (the microbial processed form of dietary choline and carnitine) and hepatic (FMO3-processed TMA) TMAO were found to be inversely correlated with steatosis [29]. In a study with 86 patients with biopsy-proven NAFLD plasma metabolites, hypoxanthine and inosine were found to be enriched in individuals with mild NAFLD. In contrast, the plasma metabolites succinate, malate, alfa-ketoglutarate, glutamine, serine and fumarate (associated with carbon metabolism) were enriched in individuals with advanced NAFLD with fibrosis [53].

Analysis of the plasma metabolome from patients with NAFLD, NASH and cirrhosis identified several metabolites as biomarkers relevant for determining disease stage. The metabolites isocitric acid and isoleucine were decreased in controls and increased with disease progression. In contrast, xanthine, glutathione and glycolic acid were found to be higher in controls and decreased with disease progression. Valine (BCAA), asparagine, propanoic acid (SCFA), butanoic acid (SCFA), phenylalanine (AAA), palmitic acid (FA), stearic acid (FA) and taurocholic acid (BA) were also identified as relevant for the metabolic signature of NAFLD; these metabolites are linked to pathways involved in bile acid, lipid, and amino acid metabolism. With increases in disease severity, taurocholic acid, phenylalanine and BCAAs increase, while glycolic acid (SCFA) and glutathione decrease [68]. Interestingly, a deficiency of serine, a precursor of glycine which is a precursor of glutathione, is associated with fatty liver disease [69,70]. In rodents, treatment with glycine improves NAFLD via glutathione synthesis [71].

SCFAs are associated with anti-inflammatory properties, in accordance with the observed decrease in NAFLD disease severity [72]. In contrast, BCAAs are associated with inflammation and insulin resistance [25,73]. Increased serum levels of BCAAs are seen in individuals with insulin resistance, and *Prevotella copri* and *Bacteroidetes vulgatus* have been identified as the main species for the association between the biosynthesis of BCAAs and insulin resistance [26]. Insulin resistance is associated with increased hepatic de novo lipogenesis, contributing to fatty liver disease [74]. Although Masarone and colleagues did not describe gut microbiome data [68], the reported metabolites such as BCAAs and AAAs, SCFAs and bile acids are in line with other studies on the gut microbiome, metabolites and NAFLD, as described in this review (Table 1).

As mentioned before, bile acids are metabolized and reabsorbed in enterohepatic circulation. Only approximately 5% of bile acids are excreted in the feces, and thus can be measured as stool metabolites. Higher serum and fecal bile acids levels are associated with advanced NAFLD and fibrosis in obese patients [75,76]. Increased serum bile acids were primary conjugated bile acids such as glycocholic acid (GCA) and secondary conjugated bile acids [75]. Increased fecal bile acids were mainly secondary unconjugated bile acids such as deoxycholic acid (DCA). Serum CGA levels and fecal DCA levels correlated with the abundance of *Bacteroidaceae* and *Lachnospiraceae* [75]. In patients with NASH, total primary bile acids were elevated and secondary bile acids were decreased [77]. In patients with fibrosis, primary bile acids were also elevated, mainly, glycine-conjugated bile acids, but secondary bile acids were the same compared to patients without fibrosis [76]. Glycogenodeoxycholic acid (GCDCA) and glycocholic acid (CGA) had strongest association with advancing fibrosis grade, as well as secondary bile acids 7-keto-deoxycholic acid (7-Keto-DCA) and glycoursodeoxycholic acid (GUDCA) [76]. Patients with borderline NASH (NAS = 3–4) had higher levels of total bile acids, total primary bile acids, and primary glycine-conjugated bile acids compared to simple steatosis (NAS = 1–2); no differences were observed between simple steatosis and definite NASH (NAS 5–8) [76]. Secondary bile acids 7-Keto-DCA and 7-Keto-lithocholic acid (7-Keto-LCA), both formed by microbial transformation, were increased in patients with definite NASH [76]. Primary bile acids, including total glycine- and taurine-conjugated bile acids, were increased in lobular inflammation and were also more increased with higher inflammation grade [76]. In steatosis, serum total cholate/chenodeoxycholate ratio is increased. Compared to steatosis or controls, in patients with NASH, total conjugated primary bile acids, conjugated/unconjugated chenodeoxycholate, cholate, and total primary bile acids are increased [77].

Different patterns of bacterial taxa-metabolites networks are observed between non-obese NAFLD and obese NAFLD [67]. In non-obese patients with worsening fibrosis severity, stool metabolites cholic acid (CA), chenodeoxycholic acid (CDCA), ursodeoxycholic acid (UDCA), glycogenodeoxycholic acid (GCDCA) and glycoursodeoxycholic acid (GUDCA) are increased. In obese individuals with significant fibrosis, LCA was significantly elevated. Of the three SCFAs, stool propionate levels gradually increased as fibrosis became more severe in non-obese patients [67].

### 3.4. Causality

Causality can be established with intervention studies. One of the first indicators for causality regarding gut microbiome and metabolic syndrome was when the transfer of intestinal microbiota of lean donors to patients with metabolic syndrome increased the insulin sensitivity [8,78,79]. Interestingly, metformin alters the gut microbiome of individuals with treatment-naive type 2 diabetes, probably contributing to the therapeutic effects of the drug [80]. However, most studies are cross-sectional and therefore cannot imply causality. A causal role for the gut microbiome and metabolites in the pathogenesis of NAFLD was indicated when donor feces of patients with hepatic steatosis induced steatosis after transplantation into germ-free mice. In addition, chronic treatment with phenylacetic acid also triggered steatosis [29]. FMT of human NAFLD gut microbiota into mice promoted the progression of NASH, by increasing the accumulation of intrahepatic B-cells, thus suggesting gut microbial-driven factors contributing to hepatic inflammation and fibrosis [81].

A recent publication studied the effect of FMT from vegan donors on histologic improvements of liver biopsies from patients with NAFLD. Although no significant histological changes were observed, positive changes in liver gene expression level were detected, as well as changes in metabolites and gut microbiome [82]. Another recently published study showed that donor FMT can improve liver stiffness in obese subjects [83]. Although promising, larger studies are needed to support these data.

## 4. Gut Microbiome and Metabolites after Bariatric Surgery and Other Weight Loss Interventions

### 4.1. Bariatric Surgery and Other Weight Loss Interventions

Bariatric surgery is a successful therapy for obesity and related comorbidities, resulting in permanent weight loss and improvements of metabolic and inflammatory status, such as insulin resistance and fatty liver disease [86,87,88,89]. The most performed types of surgery are laparoscopic Roux-en Y gastric bypass (RYGB) and laparoscopic sleeve gastrectomy (SG). Other surgical interventions include omega-loop gastric bypass (OAGB), biliopancreatic diversion, adjustable gastric banding (AdGB) and vertical banded gastroplasty (VGB).

### 4.2. Gut Microbiome in Obese Population and after Bariatric Surgery

The gut microbiome of individuals with obesity differs from that of lean individuals [90]. The gut microbiome of obese patients has increased the capacity for energy harvest [91]. Bacterial diversity and gene richness are usually decreased in patients with obesity, and microbial gene richness is inversely correlated with the severity of obesity [7]. Moreover, a small but significant negative association between microbial species alpha diversity (the species variation at individual sample level) and BMI has been reported [90].

When comparing the microbiome of lean individuals with obese individuals, the gut microbiome of obese patients was characterized by a change in relative abundance of the phyla *Firmicutes* compared to *Bacteroidetes*: *Firmicutes* were increased and *Bacteroidetes* were decreased [91]. Of the *Firmicutes,* abundance of the genera *Rikenellaceae* was increased [91]. The species *Ruminococcus gnavus, Ruminococcus torques, Ruminococcus obeum, Dorea longicatena, Dorea formicigenterans, Coprococcus comes, Lachnospiraceae bacterium, Fusobacterium ulcerans* and *Fusobacterium varium* were increased [90,92]. From the phylum *Bacteroidetes*, decreased species were *Alistipes shahii* and *Alistipes senegalensis*, as well as *Akkermansia muciniphila, Fecalibacterium prausnitzii*, and multiple species from the genus *Bacteroides*: *Bacteroides thetaiotaomicron Bacteroides uniformis, Bacterioides xylanisolvens, Bacteroides ovatus, Bacteroides intestinalis* [90,92]. A recent study found *Actinomyces odontolyticus*, *Streptococcus australis, Streptococcus thermomphilus, Collinsella aerofaciens* and *Ruminococcus torques* to be the most predictive bacterial species for obesity status [90]. The computational model found that 50% variance in body fat composition and BMI can be explained by the fecal microbiome.

After bariatric surgery, both anatomy and physiology rapidly adapt to the new physiological state [93,94]. The specific mechanisms of how bariatric surgery affects the composition of the gut microbiome, however, remains to be clarified. However, changes in diet habits, gastrointestinal anatomy, nutrients and gastrointestinal transit time, bile acid metabolism and gastrointestinal pH are probably all contributing factors in altering the gut microbiome [94]. Studies show that bariatric surgery increases microbial gene richness, although it still remains low and does not increase to the average richness seen in lean controls (Table 2) [7,95].

Several studies investigated the influence of RYGB on gut microbiome. In a study of 16 patients who underwent RYGB, the gut microbiome before and 3 months after surgery was analyzed. Before surgery, the phyla *Firmicutes* and *Actinobacteria* were more abundant and *Verrucomicrobia* was less abundant compared to lean subjects. After RYGB, the abundance of these phyla was similar to the healthy controls. Only *Proteobacteria* abundance was enriched after RYGB and lower in controls [96]. Thus, compared to before surgery, the phyla *Firmicutes, Verrucomicrobia* and *Proteobacteria* were increased, and the phylum *Actinobacteria* was decreased [96], as was *Bacteroidetes* [95]. At the genus level, *Blautia, Roseburia, Faecalibacterium* (*Firmicutes*) and *Bifidobacterium* (*Actinobacteria*) were decreased; however, these genera were still more abundant when compared to lean controls [96]. Furthermore, at the species level, *Streptococcus* spp., *Akkermansia muciniphila (Verrucomicrobia), Roseburia feces, Roseburia hominis* and *Enterococcus faecalis* were increased [96,97], and *Faecalibacterium prausnitzii* was decreased after RYGB [97,98]. *F. prausnitzii,* a butyrate producer, was associated with beneficial effects on host metabolism and negatively correlated with inflammation markers [99,100,101,102].

Studies specifically analyzing gut microbiome composition after SG found enriched abundance of the species *C. comes, D. longicatena, Clostridiales bacterium, Anaerotruncus colihominis, Akkermansia muciniphila* and *B. thetaiotaomicron* [92,103]. Relative abundance of Firmicutes, Prevotella and *Bacteroides fragilis* was decreased, and the abundance of species *Akkermansia muciniphila*, *Roseburia* spp., *Bacteroidetes*, and *Bifidobacterium* was increased. One year after SG, the abundance of phylum Actinobacteria was increased compared to baseline and three months post-operation [103].

After both RYGB and SG, the species *Escherichia coli (Proteobacteria), Klebsiella pneumoniae* (Proteobacteria) and *Haemophilus parainfluenzae (Proteobacteria)* were increased [97]. Abundance of aero-tolerant bacteria, such as *Escherichia coli* and buccal species, such as *Streptococcus* and *Veillonella* spp., was relatively higher after RYGB. In contrast, anaerobes such as *Clostridium* were more abundant after SG. *Akkermansia muciniphila* was enriched in both surgeries [97]. Functional analysis found that pathways involved in cholesterol transporters, nitrate respiration and propionate production via kinase increased in RYGB, and glutamate degradation module was more abundant in SG [97].

Long-term effects of RYGB on the gut microbial composition in seven women nine years after surgery showed enriched Gammaproteobacteria and Proteobacteria, and at the genus level, increased relative abundance of *Escherichia*, *Klebsiella* and *Pseudomonas*. Three species of Firmicutes, *Clostridium difficile*, *Clostridium hiranonis* and *Gemella sanguinis*, were lower after RYGB [104]. Functional analysis found enriched microbial genes associated with TMAO in both RYGB and ABG compared to controls, but only after RYGB increased was plasma TMAO observed.

Some patients have considerable weight regain after RYGB, and the gut microbiome may be involved. A study comparing patients at least 5 years after RYGB with and without weight regain found differences in the composition of the gut microbiome [105] In the non-regain group, higher *Akkermansia* genus abundance was found, compared both to the control group and weight-regain group. Compared to the control group, the non-regain group also had a higher abundance of *Phascolarctobacterium* genus and lower abundance of the SMB53 genus. *Bacteroidetes* still had a lower abundance after RYGB, both with and without weight regain compared to the control group, and in the control group genera *Bacteroides* and SMB53 were increased [105].

Multiple studies report changes in microbial composition after bariatric surgery. Different interventions showed different microbial profiles and only partial restoration towards lean microbiome composition. However, variations in results were observed, possibly due to different methods, small sample sizes, and/or comorbidities such as type 2 diabetes [106,107].

### 4.3. Plasma Metabolites after Bariatric Surgery

Plasma metabolites such as tyrosine, phenylalanine, leucine, isoleucine, valine and glutamate are all increased in patients with obesity [92]. Plasma glycine and glutamine levels are lower in obese patients [73]. Lower branched chain fatty acids (BCFAs) and higher BCAA levels are found in obese patients compared to controls [109]. Alterations of SCFA production are thought to play a contributing role in the development of obesity [20,110]. Colonic infusion with SCFA mixture increased fat oxidation and energy expenditure and decreased lipolysis in obese men [111]. Studies correlate metabolites with findings in the gut microbiome. For example, Liu et al. reported a decreased abundance of *Bacteroides thetaiotaomicron*, a glutamate-fermenting commensal, in Chinese patients with obesity, and this decreased abundance was inversely correlated with serum glutamate concentration [92].

As described previously, the obesity-associated gut microbiome after bariatric surgery is partially restored towards that of lean patients [92]. This phenomenon can also be observed in metabolite levels (Table 2). After both SG and RYGB, the production of AAAs and BCAAs is reduced and normalizes towards lean control levels [92]. Glycine levels relatively increase and normalize towards lean control levels [92]. In addition to the BCAAs leucine, isoleucine, and valine, the aromatic AAs phenylalanine and tyrosine, as well as ornithine, citrulline, and histidine, all decreased after RYGB. In a study comparing diet intervention with RYGB, this effect was not seen after diet intervention [108]. Furthermore, during 50 weeks of dietary weight loss, plasma SCFA levels did not change; only acetate concentration decreased with overall weight loss [112].

Correlation analysis of metabolites and clinical parameters after RYGB indicated that a cluster of metabolites, including glycine, acetylglycine, and methylmalonate, were increased and were negatively correlated with decreases in body corpulence and adipocyte diameter [7]. After RYGB and AGB, acetylglycine and glycine were negatively associated with improved weight and body composition [7]. Corresponding pathways involved in carbohydrate fermentation, the citrate cycle, glycosaminoglycan degradation, and LPS synthesis were normalized towards levels of lean controls. In patients with type 2 diabetes who underwent RYGB, plasma LPS, the endotoxin associated with insulin resistance and increased inflammatory tone, was reduced by 20% during follow-up [113]. Additionally, inflammatory stress markers decreased, and gut permeability was reduced after RYGB. Although an increase in Gram-negative, LPS-producing Proteobacteria was seen after RYGB, plasma LPS decreased, possibly due to the decreased gut permeability in combination with the increase in SCFAs, which also decreased gut permeability.

When correlated with gut microbiome composition, plasma metabolites positively associated with microbial gene richness are 3-methoxyphenylacetic acid, phloretate, hippurate, 3-hydroxphenylacetate, L-histidin, and three unidentified metabolites. Specifically, plasma glutamate is negatively correlated with microbial gene richness [7]. Changes in methylmalonate and glycine are significantly correlated with the change of *Bacteroides finegoldii* and *Coprobacillus* spp., which were also associated with improvements in body composition. Moreover, stool metabolite SCFAs acetate, propionate and butyrate were decreased after RYGB and VBG. After RYGB, the SCFA/BCFA ratio was decreased without changes in total fiber intake and with decreased protein intake, indicating that the findings were not due to dietary consumption [104].

Finally, plasma bile acids are also altered after SG and RYGB [84,114]. Especially, after RYGB, the bilio-enteric flow is altered with changes in plasma bile acid concentrations and decreases in FGF-19 [114]. Changes in bile acid profile after laparoscopic sleeve gastrectomy are associated with improvements in metabolic profile and fatty liver disease (as measured via serum cytokeratin 18) [84]. A study analyzing the gut microbiome and serum bile acids of seven patients after RYGB with controls provided evidence for an increased abundance of microbial genes involved in converting primary to secondary bile acids [104].

## 5. NAFLD after Bariatric Surgery

NAFLD improves following bariatric surgery, both histologically and as measured with non-invasive methods such as laboratory results and transient elastography (Table 3). In a prospective study of patients after SG and RYGB, changes in non-invasive measures of NAFLD before surgery and 1 year after surgery were measured. Improvements in the laboratory-based fibrosis score ASAT/ALAT ratio (0.8 ± 0.3 vs. 1.1 ± 0.4), NAFLD fibrosis score (− 1.0 ± 1.8 vs. − 1.7 ± 1.3), APRI score (0.3 ± 0.2 vs. 0.3 ± 0.1), and BARD score (2.3 ± 1.2 vs. 2.8 ± 1.1) were seen [115]. Liver function tests in biopsy-proven NAFLD improved in 84% of patients after RYGB and sleeve gastrectomy [116]. Liver stiffness measured by transient elastography improved from before surgery to post-operation (12.9 ± 10.4 vs. 7.1 ± 3.7 kPa), and RYGB showed more improvements to liver stiffness than SG [115].

Both SG and RYGB lead to histological improvements of NAFLD, as scored by NAS [117]. A prospective analysis on the effect of bariatric surgery on hepatic inflammation and fibrosis, as assessed in liver biopsies from 32 patients, found significant improvements in steatosis, lobular inflammation, ballooning, and fibrosis, and NASH was resolved in three out of four patients [118]. A recent large, retrospective, cohort analysis reported decreased progression of NAFLD to cirrhosis after bariatric surgery: 1.7% of the nonsurgical population and 0.5% of the bariatric surgery cohort had progression to cirrhosis [119]. In a retrospective analysis comparing RYGB with diet intervention on the potential effects on the severity and course of NASH, a 93.3% regression of NASH was shown after RYGB, as well as a 27.3% regression after diet [120]. In addition, bariatric surgery can be performed in selected patients with liver cirrhosis [121,122]. However, with more profound fibrosis, the long-term net clinical benefit compared to conservative weight loss therapy needs to be determined.

Bariatric surgery has positive effects on the clinical parameters of NAFLD; therefore, the pathophysiological mechanism has been studied intensively. However, studies focusing on evaluating the effect of microbiota of individuals before and after bariatric surgery on NAFLD are scarce. Whether gut microbiomes and metabolites contribute to the metabolic improvement and improvements of NAFLD seen after bariatric surgery has not been proven. For example, the plasma metabolite glycine is relatively decreased in individuals with obesity or NAFLD, and after bariatric surgery, it increases. Glycine levels are inversely correlated with BCAA levels. In rodent studies, glycine suppletion improved fatty liver disease [71], but suppletion did not lower elevated BCAA levels [123,124]. The glycine increase after bariatric surgery possibly contributes to the beneficial effect on NAFLD. Furthermore, a low-isoleucine diet in rodents reprogrammed liver and adipose metabolism, increasing hepatic insulin sensitivity and increasing energy expenditure [125]. However, further research in humans is necessary to validate hypotheses regarding the underlying metabolic processes of how bariatric surgery affects the pathophysiology of fatty liver disease.

As mentioned before, overlap of differences in microbial composition exists between different metabolic diseases, including NAFLD and diabetes. To determine the exact isolated microbial effect of bariatric surgery on NAFLD, including its bacterial metabolites, more research is necessary in human studies with large sample sizes to find the subtle differences between the metabolic diseases. Additionally, it is expected that adding machine learning bioinformatics will further increase insights into genes and pathways involved in the improvement of NASH after RYGB [126].

## 6. Discussion and Future Perspectives

Gut microbiome and plasma metabolite research is rapidly evolving, and heterogeneous new methods have been introduced. This certainly sometimes contributes to contradictory findings. Study population, chosen time points after intervention, differences in the handling of samples and storage also have an impact on study results. Validation and reproduction of results is necessary before findings can be declared as facts. Correct sample sizes with sufficient power can contribute to that. Furthermore, conflicting results and overlap of altered microbiota between metabolic diseases such as obesity, NAFLD, and type 2 diabetes are complicating factors in interpretation and in allocating findings to a distinct signature per disease: a recent review focused on microbial signatures of patients with NAFLD and the overlap with other metabolic disorders [54,67]. Thus, further research is necessary for validation and the continued comparing and interpreting of results.

Gut microbiome and metabolome research continues to evolve in two directions. First, it is evolving towards better understanding of the pathophysiology and underlying metabolic pathways of various diseases and phenotypes, using an integration of metabolome, microbiome, and other omics data [127]. For example, besides the gut microbiome, the intrahepatic microbiome and its relationship with fatty liver disease has been investigated to better understand the gut–liver axis [128,129].

Secondly, the research is evolving towards improving clinical care via determining biomarkers (serum and/or stool) for providing better methods for diagnostics, prognostics and surveillance. Furthermore, the possibilities of clinical therapies with next-generation (bacterial strain based) probiotics, albeit via fecal microbiota transplantations or oral probiotics, are explored [130]. One study showed that administration of the probiotic *Akkermansia muciniphila* reduces insulin resistance in overweight individuals [131]. A randomized controlled trial of patients with NAFLD showed that 1 year of administration of a synbiotic combination (probiotic and prebiotic) with *Bifidobacterium animalis* subspecies lactis BB-12 as a component, altered the fecal microbiome but did not reduce liver fat content or markers of liver fibrosis [132]. An RCT including patients with NAFLD scheduled for SG did not find improvements of hepatic, inflammatory or clinical outcomes when comparing the effect of generic probiotics and placebo [133]. Limitations of intervention therapies with microorganisms include the inability to control the development of the particular gut microbiome, specific abundances, or optimal ratios. Furthermore, antibiotic therapies have taught us that influencing the gut microbiome comes with adverse effects. Nowadays, the only accepted microbiome intervention to restore dysbiosis is the treatment of *Clostridium difficile* [134,135]. Developing therapies focused on specific receptors identified in pathophysiological studies could enable more specific interventions, with fewer uncontrolled influencing variables and thus less variance in preferred outcomes.

Although outside the scope of this review, animal studies can be used to further develop and evaluate hypotheses [136,137,138,139]. For example, a recent study in rats concluded that RYGB led to greater liver fat loss compared to a low-calorie diet, possibly due to increased fasting bile acid levels and the increased expression of modulators of liver fat oxidation, FXR and PPARα (peroxisome proliferator activator receptor alpha) [140]. Intervention with resveratrol altered the gut microbiota and improved hepatic steatosis and insulin resistance in mice [141]. Whether microbiota and their metabolites are a causal factor interacting with pathophysiological processes remains to be demonstrated. Intervention studies including FMT studies continue to explore the causality of the gut microbiome and derived metabolites in fatty liver disease [82,142].

Thus, the positive effects of bariatric surgery on NAFLD and its underlying mechanisms become more elucidated every year. Bioinformatics are used to exert insights into genes and pathways involved in the improvement of NASH after RYGB [46]. However, most patients with NAFLD will not undergo bariatric surgery. Therefore, expanding knowledge and the translation of gained insights is necessary to optimize general care. Although research can focus and reveal processes that cannot be seen by the human eye, we cannot escape the bigger picture of the impact of metabolic syndrome and NAFLD: there are limits to what health care can comprehend and support. The boundaries of resilience have become painfully clear during the ongoing COVID-19 pandemic: simple steatosis is not as benign as assumed [143].

## Figures and Tables

**Figure 1 metabolites-11-00353-f001:**
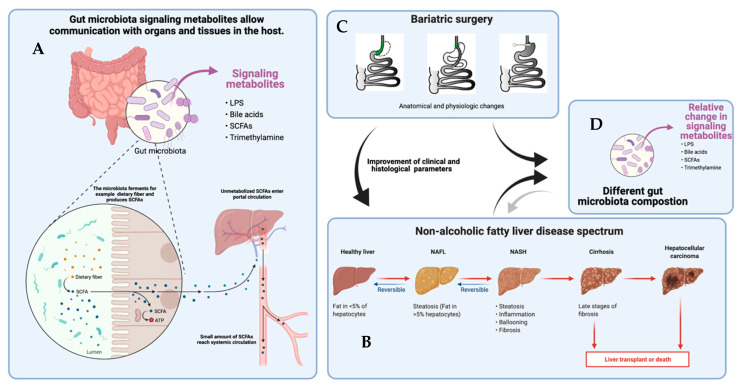
Overview of the gut microbiome and metabolites, and relationship with NAFLD and bariatric surgery. (**A**) A systematic overview of how gut microbiota contribute to producing metabolites such as LPS, bile acids, SCFAs and trimethylamine. The metabolites enter the portal circulation, where they are further metabolized and enter the systemic circulation. (**B**) The spectrum of non-alcoholic fatty liver disease. The black arrow indicates that the different stages of NAFLD are associated with a different composition of gut microbiome and plasma metabolites. The smaller grey arrow indicates the possible relationship of gut microbiome composition and plasma metabolites on non-alcoholic fatty liver disease; however, this causal role remains to be verified. (**C**) Three types of bariatric interventions are displayed: sleeve gastrectomy, Roux-en-Y gastric bypass, and vertical gastric banding. Bariatric surgery is associated with changes in gut microbiome composition and altered metabolite levels. Bariatric surgery is also associated with improvements of fatty liver disease. (**D**) Different gut microbiota compositions and relative changes in metabolites are observed in individuals with NAFLD and after bariatric surgery compared to (lean and/or pre-operative) controls. Evidence points toward a causal role for microbiome and metabolites in the pathophysiology of fatty liver disease and improvement after bariatric surgery; however, this remains to be verified. LPS, lipopolysaccharides; SCFAs, short-chain fatty acids; ATP, adenosine triphosphate; NAFLD, non-alcoholic fatty liver disease; NASH, non-alcoholic steatohepatitis.

**Table 1 metabolites-11-00353-t001:** NAFLD, gut microbiome and metabolites.

Author, Year	Study Descriptive (Population; Follow-up; Weight Loss)	Method	Microbiome	Metabolites
Belgaumkar, 2016 [84]	Prospective analysis; SG (*n* = 18) NAFLD defined by serum cytokeratin 18 (*n* = 14, 78% NAFL) FU 6 months; TWL-39.9 kg	Bile acids: LC/MS	Not described.	↑ primary glycine- and ↑ taurine-conjugated BA, ↓ cholic acid decreased, and ↑ secondary BA, ↑ glycine-conjugated urodeoxycholic acid No change in total BA.
Boursier, 2016 [64]	Biopsy-proven NAFLD (*n* = 57) F0/FI *n* = 30 vs. F3/F4 *n* = 27	Fecal microbiome: 16 S RNA sequencing analysis	Increased NAFLD severity: ↑ *Bacteroidaceae, ↓ Prevotellacea; ↓ Erysipelotrichaceae.* NASH (compared to no NASH): ↑ Bacteroides; ↓ Prevotella Significant fibrosis (F3/4) compared to F0/F1: ↑ Bacteroides; ↑ Ruminococcus; ↓ Prevotella.	Not described.
Loomba, 2017 [53]	Prospective analysis biopsy-proven NAFLD (*n* = 86): comparison mild/moderate (*n* = 72;) vs. advanced fibrosis (*n* = 14);	Fecal microbiome: whole-genome shotgun sequencing	NAFLD—mild/moderate: ↑ abundance Firmicutes; most abundant Eubacterium rectale, Bacteroides vulgates NAFLD-AF: ↑ abundance Proteobacteria; most abundant B. vulgates, Escherichia coli. ↓ Ruminococcus obeum CAG:39; R. obeum; E. rectale.	NAFLD—mild/moderate: serum: ↑ Hypoxanthine, ↑ Inosine; Stool: ↑ L-lactate; ↑ Acetate ↑ formate; NAFLD-AF: serum: ↑ Succinate; ↑ Malatae; ↑ alfa-ketoglutarate; ↑ Serine; ↑ Glutamine; ↑ Fumarate; ↑ Glutamate; ↑ Lactate; stool: ↑ butyrate, D-lactate, propionate, succinate
Caussy, 2018 [85]	Cross-sectional analysis twin family cohort, *n* = 156 validation cohort, *n* = 156 hepatic steatosis, *n* = 57	Fecal microbiome: whole-genome shotgun metagenomic sequencing; Liver: MRI/MRE; Metabolites CG/MS and LC/MS/MS	Proteobacteria, Firmicutes, Bacteroidetes correlated with 3-(4-hydroxyphenyl)lactate and phenyllactate.	6 microbial origins: 3-(4-hydroxyphenyl)lactate, N-formylmethionine, phenyllactate, mannitol, allantoine, N-(2-furoyl)glycine. 3-(4-hydroxyphenyl)lactate gut microbiome-linked metabolite assosated with liver fibrosis.
Caussy, 2019 [62]	Cross-sectional; *n* = 203 NAFLD-cirrhosis, NAFLD, without advanced fibrosis non-NAFLD controls	Fecal microbiome: 16S RNA sequencing analysis Liver: MRI/MRE.	NAFLD–cirrhosis: ↑ *Streptococcus*; ↑ *Megashaera*; most enriched abundance of family *Enterobacteriaceae*, genera *Streptococci* and *Gallibacterium*. NAFLD-AF: ↑ *Streptococcus*; ↑ *Bacillus*; ↑ *Lactococcus* Non-NAFLD: ↑ *Bacillus*; ↑ *Lactococcus*; ↑ *Pseudomonas*; ↑ *Faecalibacterium prausnitzii*, ↑ genus *Catenibacterium*; families ↑ *Rikenellaceae*, ↑ *Mogibacterium*, ↑ *Peptostreptococcaceae*	Not described.
Puri, 2018 [77]	Cross-sectional analysis biopsy-proven NAFLD and bile acids; *n* = 86 (controls *n* = 24. NAFL *n* = 25; NASH *n* = 37; BMI 31.9)	LC/MS	Not described.	NASH: ↑ total primary BAs; ↓ secondary BAs. NASH vs. NAFL, vs. controls: ↑ Total conjugated primary BAs ↑ conjugated/unconjugated chenodeoxycholate; ↑ cholate; ↑ total primary BAs. NAFL: ↑ Total cholate/chenodeoxycholate ratio ↑ total secondary/primary BA ratio -> ↓ likelihood of significant fibrosis (F ≥ 2) ↑ conjugated cholate -> ↑ likelihood of significant fibrosis (F ≥ 2).
Hoyles, 2018 [29]	Prospective analysis; obese women *n*= 105; liver biopsy (histology), NAFLD (*n* = 56); fecal microbiome (*n* =56,	Fecal microbiome: shotgun metagenomic sequencing; serum and urine Metabolites: LC/MS	Steatosis: ↑ *Proteobacteria*, ↑ *Actinobacteria*, *Verrucomicrobia* ↑ correlated *Firmicutes* and *Euryarchaeta* ↓ correlated. Species: ↑ *Acidaminococcus*, ↑ *Escherichia*; ↓ *Lachnospiraceae*, ↓ *Ruminococcaceae* Functional analysis: ↑ LPS and peptidoglycan biosynthesis.	Steatosis: Serum BCAAs: ↑ leucine, ↑ valine, ↑ isoleucine. ↑ phenylacetic acid (PAA) Urine: ↑ choline No-NAFLD: ↑ acetate; ↑ TMAO
Lee, 2020 [67]	Prospective analysis Non-obese NAFLD	Fecal metabolites: 16S RNA sequencing analysis	Elevated *Ruminococcaceae* and *Veillonellaceae* associated with fibrosis severity.	Fecal metabolites: bile acids and propionate elevated (especially with significant fibrosis).
Adams, 2020 [75]	Prospective analysis liver biopsy *n* = 122 (as part of clinical care or during bariatric surgery)	Fecal microbiome: 16S RNA sequencing analysis Metabolies: serum + fecal BA analysis: LCMS.	NAFLD-AF (F3/4): ↑ Firmicutes, ↑ *Proteobacteria*; ↑ *Actinobacteria*; ↓ *Bacteriodetes*. Family: ↑ *Actinomycetaceae*; ↑ *Lachnospiraceae*; ↓ *Bacteroidaceae*; ↓ unclassifiable of order *Bacteroidales*.	Progressive ↑ total serum BAs from controls, F0–2 NAFLD to F3/4 NAFLD. ↑ GCA (glycocholic acid); ↑ GDCA (glycodeoxycholic acid) Fecal BA: ↑ DCA (deoxycholic acid); ↑ LC (lithocholic acid).
Masarone, 2021 [68]	Cross-sectional analysis cohort biopsy-proven NAFLD *n* = 144 steatosis, *n* = 76, NASH *n* = 23, cirrhosis, *n* = 43 (NASH–cirrhosis *n* = 15, HCV *n* = 8, cryptogenic *n* = 20)	Serum metabolites GC/MS; machine learning model.	Not described.	Lower in controls and increase with disease progression: isocitric acid, isoleucine, not identified metabolite Higher in controls and decrease with disease progression: xanthine, glutathione, glycolic acid Valine, asparagine, 4-deoxy erythronic acid, propanoic acid, palmitic acid, butanoic acid, stearic acid, phenylalanine, taurocholic acid NASH-related cirrhosis, increased concentration of galactose, uric acid, glyceric acid, butanoic acid, histidine, phenylalanine, stearic acid, threonine and palmitic acid
Nimer, 2021 [76]	Prospective analysis; NAFLD *n* = 102 (30% simple steatosis, 43% borderline NASH, 27% NASH); controls *n* = 50 Liver biopsy; BMI 32.8 kg/m2	Plasma bile acid profile: quantitative stable isotope dilution LC/MS/MS	Not described.	NAFLD vs. controls: ↑ almost all circulating BAs Fibrosis vs. NAFLD: ↑ glycine-conjugated primary BAs (↑ GCDCA, ↑ GCA), secondary BAs ↑ 7-keto-DCA, ↑ GUDCA NASH vs. simple steatosis: ↑ 7-keto-DCA, ↑ 7-keto, LCA

**Table 2 metabolites-11-00353-t002:** Bariatric surgery, gut microbiome and metabolites.

First author, year [Ref.]	Study Descriptive (Population; Follow-up; Weight Loss)	Method	Results Microbiome	Results Metabolites
Laferrere, 2011 [108]	Prospective analysis; diabetic patients; RYGB, *n* = 10 Diet intervention, *n* = 11 FU 1 month	Plasma metabolites: MS	Not described.	RYGB: ↓ BCAAs: ↓ leucine, ↓ isoleucine, ↓ valine; ↓ aromatic AAs: ↓ phenylalanine ↓ tyrosine; ↓ ornithine, ↓ citrulline, ↓ histidine. Diet: no differences
Tremaroli, 2015 [104]	Post-bariatric surgery, long-term effect: RYGB (*n* = 7) vs. VGB (*n* = 7); matched controls (severe obesity *n* = 7); FU 9.4 years	Fecal microbiome: shotgun metagenomic sequencing Metabolites: GC/MS, UPLC-MS-MS	RYGB vs. obese controls: ↑ *Gammaproteobacteria*; ↓ *Firmicutes* (↓ *Clostridium difficile*, ↓ *Clostridium hiranonis*, ↓ *Gemella sanguinis*); ↑ *Proteobacteria* (↑ *Escherichia*, ↑ *Klebsiella*, ↑ *Pseudomonas*) No significant differences VBG and controls. No significant differences VBG and RYGB.	RYGB, VBG: ↓SCFAs (↓ acetate, ↓ propionate, ↓ butyrate) BCFA (isobutryrate, isovaleratie): no change. RYGB: SCFA/BCFA ratio decreased
Palleja, 2016 [98]	Prospective analysis; RYGB, *n* = 13 FU 3 months, *n* = 12 12 months, *n* = 8	Fecal microbiome: Shotgun metagenomic sequencing	Microbial diversity: ↑ 3 months and ↑ 1 year. ↑ Proteobacteria, ↑ Fusobacteria ↑ *Escherichia coli*, ↑ *Klebsiella pneumoniae*, ↑ *Veillonella spp*., ↑ *Streptococcus spp*., ↑ *Alistipes spp*., and ↑ *Enterococcus faecalis*; ↑ *Bifidobacterium dentium*; ↑ *Fusobacterium nucleatum*; ↑ *Akkermansia muciniphila*; ↓ *Faecalibacterium prausnitzii*	Not described.
Liu, 2017 [92]	Prospective analysisSG, *n* = 23 obese individuals, *n* = 72 lean controls, *n* = 79 FU 1 and 3 months.	Shotgun metagenomic sequencing; serum plasma metabolites: LC/MS	↑ : *C. comes; D. longicatena; Clostridiales bacterium; Anaerotruncus colihominis; Akkermansia muciniphila; B. thetaiotaomicron.*	↓ : Aromatic amino acids ↓; methionine ↓; alanine ↓; lysine ↓; serine ↓; glutamate ↓ decreased ↑ : Acetylglycine ↑; glycine ↑increased.
Aron, 2019 [7)]	Prospective analysisRYGB, *n* = 41 gastric banding/AGB, *n* = 20) *n* = 24 had post-operative follow-up.	Fecal microbiome: shotgun metagenomic sequencing Serum metabolites: LC/MS	RYGB: ↑ *Oscillibacter*; ↑ *Clostridium* sp; ↑ *Alistipes shahii*; ↑ *Butyricimonas*; ↑ *Fusobacterium nucleatum*; ↑ *Roseburia*; ↑ *Dialister sp*; ↑ *Butyricimonas virosa*; ↑ *Hungatella hatewayi*; ↓ *Coprobacillus* sp.; ↓ *Anaerostipes hadrus* (butyrate producer)AGB: ↑ *Oscillibacter*; ↑ *Butyricimonas virosa*; ↑ *Bacteroides finegoldii*	Metabolites associated with microbial gene richness (MGR): glutamate, negatively correlated; 3-methoxyphenylacetic acid, phloretate, hippurate, 3-hydroxphenylacetate, L-histidin and three unidentified positively correlated.. RYGB: ↑ glycine, ↑ acetylglycine, ↑ methylmalonate After BS: - ↑ Acetlyglycine and ↑ glycine: - ↓ BCAA when insulin resistance ↑
Steinert, 2020 [96]	Prospective *n*= 25, RYGB, *n* = 16 (RYGB), Controls, *n*= 9 FU 3 months, not described	Fecal microbiome: 16S RNA sequencing analysis	RYGB vs. pre-operation: ↓ *Blautia*, ↓ *Roseburia* and ↓ *Faecalibacterium* (Firmicutes); compared to controls more abundant RYGB: ↓ *Bifidobacterium*, which was increased compared to lean controls	Not described.
Farin, 2020 [97]	RYGB (*n* = 89) SG (*n*= 108) FU 6 months; not described	Fecal microbiome: shotgun metagenomic sequencing	*Phylum*: 15% gene enrichment: ↑ *Bacteroidetes*, ↑ Proteobacteria, ↑ *Actinobacteria*, ↑ *Verrucomicrobia*; ↓ *Firmicutes* (*Firmicutes* slight decrease after surgeries).Both: ↑ *A*. *muciniphila*; ↑ *E. coli*, ↑ *H*. *parainfluenzae*, ↑ *Klebsiella pneuoniae*. RYGB > SG: ↑ *E*. *coli* and ↑ K. *pneumoniae*, ↑ *R*. *faecis* and ↑ *R. hominis*, ↑ *E. faecalis* more enriched by RYGB than SG. ↓ *Faecalibacterium preusnitzii* less abundant after RYGB. SG > RYGB: ↑ *A. hadrus*, C. sp KLE, F. *plautii*, O. sp. KLE, *R*. *gnavus* (*Firmicutes* order *Clostridiales*)	Not described.
Karami, 2020 [95]	Prospective analysis *n* = 30 RYGB, *n* = 12 SG, *n* = 18 FU 6 months; EWL 57.72 ± 15.08%	Fecal microbiome: 16S RNA sequencing analysis	RYGB: *Bacteroidetes* ↓ compared to pre-surgery. RYGB or SG: no changes in *Firmicutes* count. *Firmicutes* to *Bacteroidetes* ratio not different from baseline; *B*. *fragilis* count not different to baseline or between groups. BS (RYGB + SG): ↓ *Bacteroidetes*; ↑ *Firmicutes*/*Bacteroidetes* ratio.	Not described.
Faria, 2020 [105]	Retrospective analysis; RYGB: with vs. without weight regain, vs. control; FU > 5 year post-operative; TWL% 25.8 vs. 33.1.	Fecal microbiome: 16S RNA sequencing analysis.	RYGB, non-regain vs. control and weight regain: ↑ *Akkermansia* genus. RYGB, non-regain vs. control: ↑ *Phascolarctobacterium* genus and ↓ SMB53 genus. RYGB vs. control: ↓ *Bacteroidetes* Control vs. RYGB: ↑ *Bacteroides,* ↑ SMB53	Not described.
Pakiet, 2020 [109]	Prospective analysis OAGB, *n* = 50 lean controls, *n* = 32	Serum metabolites GM/LM/MS	Not described.	Baseline vs. controls: ↓BCFAs; ↑ BCAAs OAGB: ↑ BCFA; ↓BCAA. IR correlated inversely with BCFAs and positively with BCAAs
Tabasi, 2021 [103]	Prospective analysis SG, *n* = 126 FU 3 and 12 months	Fecal microbiome: qPCR	M3: ↓ *Prevotella*; ↓ *Bacteroides fragilis* group; ↓ *Firmicutes* spp; ↑ *Akkermansia muciniphila*; ↑ *Roseburia* spp.; ↑ *Bacteroidetes*; ↑ *Bifidobacterium* M12: ↑ *Actinobacteria* (compared to M3 and baseline)	Not described.

**Table 3 metabolites-11-00353-t003:** Bariatric surgery and NAFLD.

Author, Year [Ref.]	Study Design	Parameters to Assess Liver Disease	Changes/Outcomes
Nickel, 2018 [115]	Prospective analysis, *n* = 100 SG vs. RYGB FU 1 year.	Transient elastography: liver stiffness Laboratory-based fibrosis score: AST/ALT ratio; NAFLD fibrosis score; APRI score; BARD score	12.9 ± 10.4 vs. 7.1 ± 3.7 kPa; (RYGB > SG) 0.8 ± 0.3 vs. 1.1 ± 0.4 −1.0 ± 1.8 vs. −1.7 ± 1.3; 0.3 ± 0.2 vs. 0.3 ± 0.1; 2.3 ± 1.2 vs. 2.8 ± 1.1.
Garg, 2018 [118]	Prospective analysis Bariatric surgery population *n* = 76 FU 1 year (32 biopsies)	Controlled attenuation parameter (CAP) Transient elastography: liver stiffness Liver biopsy	Pre-operation: 326.5 (301–360.5) dB/m; Success rate pre-operation 87.9% Pre-operation 7.0 (5.0–9.5) kPa Improvement in hepatic steatosis, lobular inflammation, ballooning and fibrosis; NASH resolved in 3 out of 4 patients.
Cherla, 2020 [116]	Prospective analysis of biopsy-proven NAFLD SG, *n* = 62 RYGB, *n* = 248 FU 4 years (median)	Liver function tests AST, SG AST, RYGB ALT, SG ALT, RYGB	84% normalized after bariatric surgery 49.1 ± 21.5 to 28.0 ± 16.5 49.3 ± 22.0 to 26.5 ± 15.5 61.7 ± 30.0 to 27.2 ± 21.5 59.4 ± 24.9 to 26.1 ± 19.2
Wirth, 2020 [119]	Retrospective analysis; Bariatric surgery, *n* = 2942 Control, *n* = 5884 FU 31 months	Risk of NAFLD progression to cirrhosis after bariatric surgery	Reduced risk: HR 0.31 (95% CI 0.19–0.52)

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
