# Peer review of "Gut Microbiome and Metabolites in Patients with NAFLD and after Bariatric Surgery: A Comprehensive Review"

_metabolites, 2021, doi:10.3390/metabo11060353_

Round 1

Reviewer 1 Report

In the current study, Hoozemans et al. review literature on the gut microbiome composition and related metabolites in NAFLD and bariatric surgery. The review addresses an important topic, it is interesting and relatively easy to read. There are however a number of important issues that deserve further attention. Some general and specific comments and suggestions are provided below.  

General comments:

A number of important discoveries in the field are not covered in the current work. Key concepts that may link the microbiome and related metabolites with obesity and NAFLD are under-developed. These include pathways regulating the metabolism of bile acids, specific amino acids (glycine, serine and glutathione) and alcohol-producing bacteria. Also, the concept of intrahepatic microbiome is not discussed. The review should be expanded to cover these topics. The model (Figure 1) and the tables should be revised and updated accordingly. Further details and suggestions are provided as specific comments below.

Throughout the manuscript, the authors often cite review articles instead of original work. For example: “The main SCFAs are propionate, butyrate and acetate and are associated with increased insulin sensitivity and inflammation (14)”.  “Elevated TMAO has been associated with atherosclerotic vascular disease (15)”. The original research articles should be cited throughout the manuscript. Also, references necessary to support specific statements are often missing.

Specific comments:

Abstract and Introduction: “Currently, the most effective therapy to prevent progression of NAFLD… is bariatric surgery”. This is a strong statement. According to PMID: 28714183 (Ref. 4), “Nonsurgical weight loss is effective in improving all histological features of NAFLD…”, but “it is premature to consider foregut bariatric surgery as an established option to specifically treat NASH”. Also, “bariatric surgery may be considered on a case-by-case basis…”. Therefore, statements regarding high efficacy should be toned down unless specific evidence, including comparison to other approaches, is provided.  

Section 1:

Page 2, line 45: Please provide references indicating a casual role for the microbiome in the pathogenesis of obesity, diabetes and atherosclerosis.

Page 2, Figure 1: The model is presented early, before the main concepts, terms and abbreviations are introduced. A detailed legend should be provided. The authors may consider presenting the model after the main ideas are discussed, at the end of the manuscript .  

Section 2:

Page 3, lines 85-88: How are SCFA associated with insulin sensitivity and inflammation? Please explain and provide direction of effect.

Page 3, line 88: Please correct ‘aromatase amino acids’ to aromatic amino acids.

Page 3, lines 88-99:  Please add references supporting the dysregulated metabolism of AAA and BCAA in NAFLD (e.g., PMID: 29942096 and/or others).

Section 3:

Page 4, line 139-140: The authors discuss amino acids that are increased in obesity, but do not cover amino acids that are decreased in obesity and are normalized after weight loss and/or bariatric surgery. For instance, glycine and serine are increased after sleeve gastrectomy (PMID: 28628112, Ref. 22) and these two amino acids play important roles in NAFLD/NASH. Furthermore, glycine is known to be lower not only in obesity, but also in NAFLD, diabetes and in cardiovascular diseases. Recently, glycine-based treatment was shown to reduce NAFLD in mouse models and to modulate glutathione metabolism and the gut microbiome. The gut microbiome was shown to modulate host amino acid and glutathione metabolism (PMID: 26475342, PMID: 33268508, PMID: 28802074, PMID: 28254760, PMID: 19356713, PMID: 24419221, and others). Can the authors discuss a potential role of the microbiome in glycine and serine (and glutathione) metabolism in relation to obesity (Section 3), NAFLD (Section 5) and the influence of bariatric surgery (Section 6)?

Page 4, line 140: Please define BCFA.

Section 5:

Can the authors address and offer potential explanations for some reported differences in the microbiome signature in obesity and bariatric surgery vs. NAFLD? For instance, changes in the phyla Bacteroidetes and Firmicutes in obesity were reported to be opposite in NAFLD (PMID: 31255652, PMID: 23055155). Are changes in Proteobacteria consistent or also differ in obesity and NAFLD? Considering the significant contribution of obesity to NAFLD development, what could be the reason for such differences?

Please elaborate on the changes in bile acids in NAFLD progression and the potential role of the microbiome: 1) include new literature (e.g., PMID: 33275980, and others) and 2) discuss specific members of the gut microbiota and/or mechanisms that could drive altered bile acid metabolism in NAFLD.    

As mentioned above, the discussion of the specific amino acids, glycine and serine (and glutathione) in NAFLD and the potential role of the microbiome could be expanded in this section.   

Another interesting metabolic pathway in NAFLD that was demonstrated to be regulated by the microbiome was not covered in the current work. The high alcohol-producing Klebsiella pneumoniae was recently reported to be increased in patients with NAFLD and to accelerate the disease in mice. Interestingly, as mentioned by the authors (Ref. 60 and 61), Klebsiella pneumoniae is increased after bariatric surgeries. This topic should be introduced and discussed in the context of obesity (Section 3) NAFLD (Section 5) and bariatric surgery (Section 6).

The concept of intrahepatic microbiome, its relationship with the gut microbiome and the gut-liver axis in the context of NAFLD and obesity should be introduced and discussed (e.g., PMID: 31900291, PMID: 32060128 and others).

Page 4 line 190: Please provide refences for the statement: “Functional analysis of these microbiome differences reported increased microbial capacity for metabolism of BCAAs and AAAs”.

Page 4, lines 244-246: References are missing to support statements on BCAA and SCFA.

Table 1: 1) Caussy et al. are cited twice (2018 and 2019), but found only once in the reference list (2019). Also, please provide the number of each reference in the table. 2) Please separate the study design and the methodological approaches (e.g., 16S, LC-MS, etc.) used in each study to two columns.  Also, for some references the methodological approaches were not described (Puri et al. 2018). 3) The comments above related to a better discussion of amino acids and bile acids in NAFLD could be incorporated in the revised tables.    

Section 6:

What are the similarities and the differences in microbiome alterations as well as NAFLD outcomes with respect to the different types of bariatric surgeries described in page 9, lines 290-293?

Can the authors provide any potential explanations why the phylum Proteobacteria, which is consistently reported to be higher in NAFLD patients, is found to increase after bariatric surgery?

Page 10, line 306: ‘controls’ mean lean subjects?

Page 10, line 314-315: Please add references to support the claims on F. prausnitzii.

Page 14, line 381-382: Improvement of NAFLD cannot be concluded based only on serum cytokeratin 18. Please revise.

Table 2: 1) Please correct Lui 2017 to Liu 2017. 2) Please add reference numbers and separate the study design and methodology to different columns (as mentioned above for Table 1).

Section 7:

Are there any studies assessing changes in NAFLD, the microbiome and their relationship before and after bariatric surgery? Any studies evaluating the effects of microbiota isolated from patients before and after bariatric surgery on NAFLD in mouse models? Is so, please discuss relevant literature. If not, please highlight the gap of knowledge and the need for further research.   

Reviewer 2 Report

I found the topic of this paper very interesting and relevant; data are nicely and logically presented, and the paper is overall well written.

I do not have any major complaints.

Minor complaints:

Figure 1 is not self-explanatory, and it lacks short description in the Figure legend. It is important to provide schematic summary. I suggest Figure 1 to be rebuild.

Author Response

Rebuttal Metabolites 1199589; Gut microbiome and metabolites in patients with NAFLD and after bariatric surgery: a comprehensive review

Amsterdam 25th May 2021

Dear editor,

Please find herewith our general answers to the questions of the reviewers.  Upon their suggestion, we made some general changes to the manuscript.  Moreover, there are now 6 sections: 1) Introduction, 2) Microbiome and Metabolites, 3) Gut Microbiome and Metabolites in NAFLD, 4) Gut Microbiome and Metabolites after Bariatric Surgery and other Weight Loss Interventions, 5) NAFLD after Bariatric Surgery and 6) Discussion and Future Perspectives.  Finally, we’ve incorporated section 3 (Microbiome, Metabolites and Obesity) and section 4 (Microbiome, Metabolites and Diabetes) in Section 2 (Microbiome and Metabolites), Section 3 (Microbiome, Metabolites and NAFLD) and Section 4 (Microbiome, Metabolites and Bariatric Surgery).

Furthermore, we have updated Figure 1 and provided a legenda. Figure 1 is moved toward the end of the manuscript, between section 5 and 6.  Below are our itimized responses on the detailed comments. We hope the manuscript is now suitable for publication.

Kind regards on behalf of all authors,
Jacqueline Hoozemans MD PhD student

Reviewer 3 Report

Gut microbiome and metabolites in patients with NAFLD and after bariatric surgery: a comprehensive review

In this review, the authors described the actual comprehension of the link between gut microbiota and derived metabolites for NAFLD patients and after bariatric surgery, considering the link between these two metabolic statuses. The manuscript addressed an original/specific topic. In global, the topic is very well described and the whole manuscript is very well structure and easy to read.

Comments and Suggestions for Authors

  1. General comment: Maybe it will be easier to read if NAFLD was repeated less in some section (especially in page 5, which is mentioned about 23 times in 1 page). I suggest some reformulations.
  2. Section 4: This whole section seems unnecessary for the understanding of the topic.
  3. Section 5: Considering the association between obesity and NAFLD, it will be interesting to mention if the compared groups in mention studies have different adiposity/anthropometric parameters.
  4. Section 5.2: There is some literature about the role of bacteria mentioned in this section? If so, it will be interesting to describe it briefly.
  5. Table 2. The name of column in this table should be harmonized with Table 1.
  6. Line 86: “and” is missing between pH and intestinal.
  7. Line 79: “its’” should be “it’s”?
  8. Line 106-109: Reference missing?
  9. Line 314-315: Reference missing?

Reviewer 4 Report

I enjoyed reading this nice review by Jacqueline Hoozemans and colleagues. My comments and suggestions for consideration:

  1. To be more focused on the topic of this review, I would suggest removing current sections 3 and 4 regarding microbiome and metabolites in obesity and diabetes or breaking them down and integrating into current sections 5 and 6 regarding microbiome and metabolites in NAFLD and after bariatric surgery. I found current sections 3 and 4 are not comprehensive enough for the topics of their own but a bit distracting from the main topics.
  2. Ref 15: there are better references to support the sentence, such as https://www.ahajournals.org/doi/pdf/10.1161/JAHA.116.004947. Please add citations to support the LPS studies (line 100).
  3. Current section 3.2 is underdeveloped. There are many large-scale studies reporting obesity-related metabolites, as reviewed here: https://link.springer.com/article/10.1007/s11306-019-1553-y . Meanwhile, see my first comment.
  4. Line 148: “The role of microbiome and metabolites in insulin resistance has been studied intensely”. I would tune down this sentence. As most existing studies on microbiome and insulin resistance/diabetes have been cross-sectional (can’t imply causality), with small sample sizes and limited generalizability, and usually not well controlled the confounding, I think current evidence is still suggestive.
  5. Line 248: “This supports the contributing role of gut microbiome and metabolites in the pathophysiologic mechanism behind NAFLD”. Like my last comment, I think the evidence so far suggests the bi-directional relationship of microbiome and NAFLD. The sentence does not reflect the studies cited in the review, which were mostly cross-sectional and can’t imply the causality or direction of the relationship.
  6. Please add reference numbers to Tables 1-3. And I wonder in what order the papers in the Tables were arranged, by year, author name?
  7. Some more papers to be added to Table 2: e.g., https://pubmed.ncbi.nlm.nih.gov/32747219/ , https://doi.org/10.1038/tpj.2012.43, https://doi.org/10.1155/2015/806248, https://doi.org/10.1007/s11695-016-2399-2, https://doi.org/10.1038/s41522-020-0122-5  . Please search the literature again to make sure no significant missing.
